# ENHANCE-A-VIDEO: BETTER GENERATED VIDEO FOR FREE

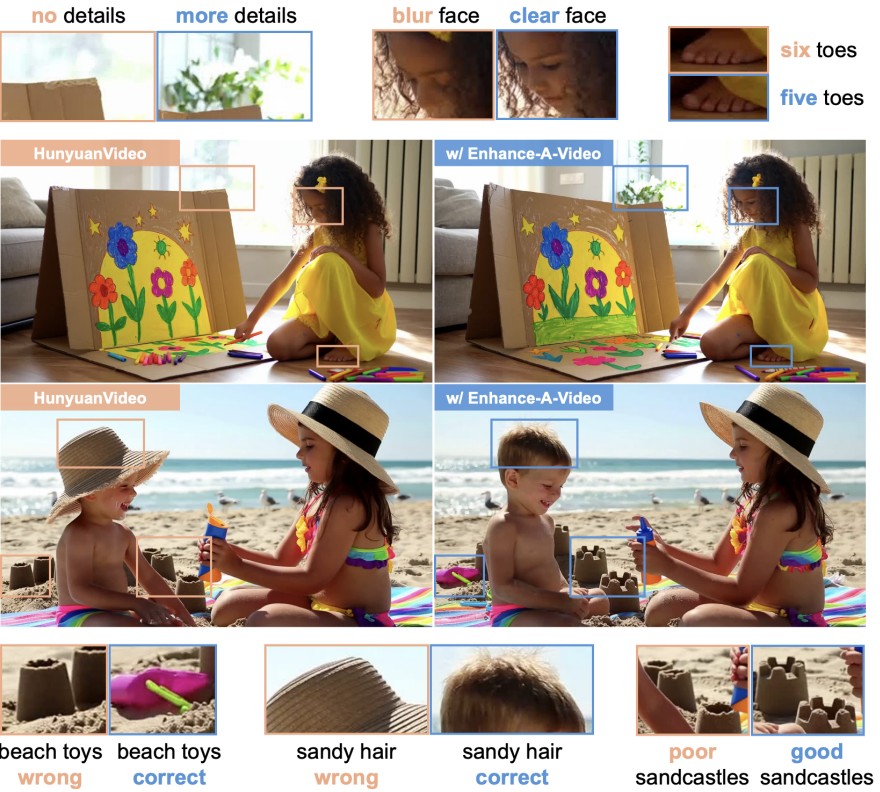

Figure 1: Enhance-A-Video boosts diffusion transformers-based video generation quality at minimal cost - no training needed, no extra learnable parameters, no memory overhead. Detailed captions are available in Appendix G.

## ABSTRACT

DiT-based video generation has achieved remarkable results, but research into enhancing existing models remains relatively unexplored. In this work, we introduce a training-free approach to enhance the coherence and quality of DiT-based generated videos, named **Enhance-A-Video**. The core idea is to enhance the cross-frame correlations based on non-diagonal temporal attention distributions. Thanks to its simple design, our approach can be easily applied to most DiT-based video generation frameworks without any retraining or fine-tuning. Across various DiT-based video generation models, our approach demonstrates promising improvements in both temporal consistency and visual quality. We hope this research can inspire future explorations in video generation enhancement.

## 1 INTRODUCTION

Diffusion transformer (DiT) models (Peebles & Xie, 2022) have revolutionized video generation, enabling the creation of realistic and compelling videos (Yang et al., 2024; Brooks et al., 2024; Lin

et al., 2024; Xu et al., 2024a; Kong et al., 2025; Wan et al., 2025). However, achieving temporal consistency across frames while maintaining fine-grained details remains a significant challenge. Many existing methods generate videos that suffer from unnatural transitions and degraded quality, as illustrated in Fig. 2, which limits their practical applicability in real-world scenarios and professional applications (Yan et al., 2023; Henschel et al., 2024).

Video generation enhancement (He et al., 2024; Ma et al., 2025) is designed for addressing the above limitations, where two objectives are primarily considered: (*i*) maintaining temporal consistency across frames, which ensures smooth and coherent transitions, and (*ii*) improving spatial details, which enhances the visual quality of each frame for more realistic video outputs. In UNet-based video generation (Zhang et al., 2023a; Guo et al., 2024; Xu et al., 2024b; Zhang et al., 2024; Xia et al., 2024; Bu et al., 2025; Yuan et al., 2025; Li et al., 2025b), Upscale-A-Video (Zhou et al., 2024) integrated a local-global temporal strategy for better temporal coherence, and VEnhancer (He et al., 2024) designed a video ControlNet (Zhang et al., 2023b) to enhance spatial and temporal resolution si-

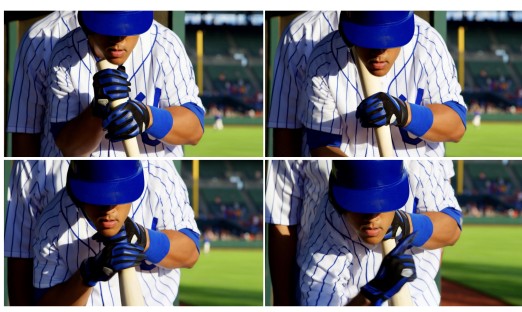

*A baseball player grips a bat in* black *gloves, wearing a blue-and-white uniform and cap, with a blurred crowd and green field highlighting his focused stance.*

Figure 2: Video sample of HunyuanVideo model with *unnatural head movements*, *repeated right hands* and *conflicting glove color.*

multaneously. Nevertheless, the exploration of enhancing DiT-based video generation remains limited, particularly in addressing challenges of temporal consistency and spatial detail preservation.

In DiT-based video generation, temporal attention (Singer et al., 2022; Villegas et al., 2023; Tan et al., 2023) plays a crucial role in ensuring coherence among frames. Through careful analysis of temporal attention in DiT blocks, we made an important observation as shown in Fig. 3: In DiT-based models, temporal attention is typically concentrated along the diagonal, enabling more efficient utilization of model capacity to capture the most relevant temporal dependencies (Lu et al., 2024; Cai et al., 2024; Li et al., 2025c). However, the imbalance between cross-frame (non-diagonal elements) and intra-frame attention (diagonal elements) may cause foundation models to underutilize global cross-frame information, resulting in inconsistencies across frames – such as abrupt transitions or blurred details in the generated videos.

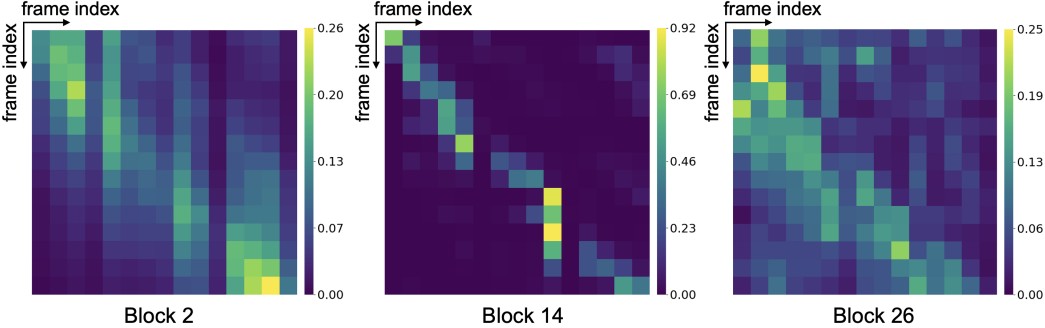

Figure 3: Visualization of temporal attention distributions in Open-Sora for blocks 2, 14, and 26 at denoising step 30, where non-diagonal elements are considerably weaker than diagonal elements.

A straightforward approach to address the highly uneven distribution between intra-frame and cross-frame attention is to incorporate the temperature mechanism (Peeperkorn et al., 2024; Renze & Guven, 2024) into the computation of temporal attention. However, this naive strategy significantly alters the original temporal attention patterns as the depth of DiT blocks and the number of denoising steps increase. Preserving the original temporal attention patterns is important, as they encode prior knowledge acquired during training for video generation. Disrupting these patterns compromises the learned knowledge and leads to reduced generation quality, as demonstrated in Appendix C.

To more effectively integrate temperature-based adjustments into temporal attention, we propose applying a cross-frame intensity-driven temperature in the residual connection (He et al., 2015).

Specifically, the cross-frame intensity is calculated as the average of the non-diagonal temporal attention weights. This value is then used to adaptively balance dependencies across frames, leading to improved video quality. Furthermore, because attention outputs contribute relatively low compared to hidden states within the residual connection (Si et al., 2024; Ma et al., 2024a), a moderate increase of the original attention output using cross-frame intensity has a limited impact on the overall attention pattern in the final representation of each DiT block.

Building on these insights, we develop a novel, training-free, and plug-and-play approach, Enhance-A-Video, to improve the temporal and spatial quality of DiT-based generated videos. The method introduces two key innovations: a cross-frame intensity to capture cross-frame information within the residual connection and an enhance temperature parameter to scale the calculated cross-frame intensity. By strengthening cross-frame correlations from the temperature perspective, our approach enhances temporal consistency and preserves fine visual details effectively. **A notable advantage is that this method can be readily integrated into prevalent DiT-based video generation frameworks with negligible computational overhead.**

We conduct a comprehensive experimental evaluation of our approach across several benchmark DiT-based video generation models, including Wan (Wan et al., 2025), HunyuanVideo (Kong et al., 2025), CogVideoX (Yang et al., 2024), LTX-Video (HaCohen et al., 2024), Open-Sora (Zheng et al., 2024) and Open-Sora-Plan (Lin et al., 2024). By incorporating Enhance-A-Video during the inference phase, these models demonstrate a significant improvement in generated video quality by reducing temporal inconsistencies and refining visual fidelity. In particular, the enhanced cross-frame attention not only mitigates temporal inconsistencies by encouraging the model to exploit richer contextual information, but also facilitates smoother transmission of prompt and spatial information across frames, thereby enabling the generation of finer visual details and more prompt-consistent videos, as illustrated in Fig. 1.

## 2 RELATED WORK

**Video Generation.** Recent advancements in video generation have been driven by powerful diffusion transformer-based models (Chen et al., 2024; Ma et al., 2024b; Gao et al., 2024; Lu et al., 2024). Sora (Brooks et al., 2024) has demonstrated exceptional capabilities in generating realistic and long-duration videos, establishing itself as a significant milestone in text-to-video generation. CogVideoX (Yang et al., 2024) introduced a 3D full attention mechanism and expert transformers to improve motion consistency and semantic alignment. HunyuanVideo (Kong et al., 2025) introduces a hybrid stream block with enhanced semantic understanding. Nevertheless, key challenges remain in video generation, including temporal inconsistency and the degradation of fine-grained spatial details.

**Temperature Parameter.** The temperature parameter is a well-known concept in deep learning, primarily used to control the distribution of attention or output probabilities in generative models (Peeperkorn et al., 2024; Renze & Guven, 2024). In natural language generation tasks, the temperature is often adjusted during inference to modulate the diversity of the generated text (Holtzman et al., 2020). A higher temperature increases randomness, promoting creativity, while a lower temperature encourages deterministic and coherent outputs. Recently, the concept has been explored in vision-related tasks, such as visual question answering and multimodal learning (Chen et al., 2021), where temperature adjustments are applied to balance multimodal attention distributions. Yet, its application within DiT-based video generation, and in particular its impact on temporal attention, has not been thoroughly investigated.

## 3 METHODOLOGY

### 3.1 DIFFUSION TRANSFORMER MODELS

Diffusion Transformer models are inspired by the success of diffusion models in generating high-quality images and videos by iteratively refining noisy data (Ho et al., 2022b;a; Xu et al., 2023; Blattmann et al., 2023; Wang et al., 2023; Esser et al., 2024). These models combine the strengths of diffusion processes and transformer architectures to model temporal and spatial dependencies in video generation. The forward diffusion process adds noise to the data over $T$ timesteps, gradually converting it into a noise distribution. Starting from clean data $\mathbf{x}_0$, the noisy data at timestep $t$ is

obtained as:

$$\mathbf{x}_t = \sqrt{\alpha_t}\mathbf{x}_{t-1} + \sqrt{1-\alpha_t}\mathbf{z}_t, \quad \text{for } t = 1, \dots, T, \tag{1}$$

where $\alpha_t$ controls the noise schedule and $\mathbf{z}_t \sim \mathcal{N}(0, \mathbf{I})$ is Gaussian noise. As $t$ increases, $\mathbf{x}_t$ approaches a standard normal distribution $\mathcal{N}(0, \mathbf{I})$. To recover the original data distribution, the reverse diffusion process progressively removes noise from $\mathbf{x}_t$ until reaching $\mathbf{x}_0$:

$$p_\theta(\mathbf{x}_{t-1}|\mathbf{x}_t) = \mathcal{N}(\mathbf{x}_{t-1}; \mu_\theta(\mathbf{x}_t, t), \Sigma_\theta(\mathbf{x}_t, t)), \tag{2}$$

where $\mu_\theta$ and $\Sigma_\theta$ are learned parameters representing the mean and covariance of the denoised distribution.

## 3.2 TEMPERATURE IN DiT-BASED VIDEO GENERATION

The temperature is a critical concept in large language model inference, controlling the randomness and coherence of the generated tokens. The probability $P(x)$ of generating a token $x$ is adjusted using the temperature $\tau$ as:

$$P(x) = \frac{\exp\left(\frac{z(x)}{\tau}\right)}{\sum_{x'} \exp\left(\frac{z(x')}{\tau}\right)} \tag{3}$$

where $z(x)$ represents the unnormalized logit for token $x$, and $\tau > 0$ controls the degree of randomness: a lower $\tau$ makes the output more deterministic, while a higher $\tau$ increases diversity by flattening the probability distribution.

In video generation, a similar temperature principle can be considered when using DiT models, where the temporal attention mechanism controls the relationship between generated frames. Eq. 4 presents a direct usage of temperature in temporal attention of DiT models. A higher $\tau$ yields a more uniform temporal attention, enabling broader context integration.

$$\text{Attn}(Q, K) = \text{softmax}\left(\frac{QK^\top}{\tau \cdot \sqrt{d_k}}\right) \tag{4}$$

Nevertheless, as illustrated in Appendix C, directly applying $\tau$ to temporal attention causes increasing changes to the original attention weights as the model deepens and denoising steps accumulate, which can lead to overly smooth motion, loss of visual details, and unstable video generation.

Therefore, incorporating a temperature into the attention output within the residual connection is more practical, given the significant magnitude difference presented in Appendix D between attention outputs and hidden states. Furthermore, non-diagonal temporal attention weights can be used to quantify cross-frame correlations, enabling adaptive temperature adjustment to enhance global information aggregation across frames, thereby improving spatial diversity and temporal consistency.

## 3.3 ENHANCE BLOCK

To better adaptively adjust the temperature in the temporal attention mechanism, we propose a novel method, Enhance-A-Video, to enhance temporal consistency in video generation by utilizing the **non-diagonal temporal attention** with **enhance temperature parameter**. The cross-frame intensity is measured by the non-diagonal temporal attention, where higher values enable the model to focus on a broader temporal context, corresponding to higher temperature. By further introducing the enhance temperature parameter to scale the cross-frame intensity, we appropriately adjust the temporal attention outputs as a training-free enhancement.

As presented in Fig. 4, we design an **Enhance Block** as a parallel branch to the temporal attention mechanism. The Enhance Block operates as follows:

First, the temporal attention map $A^{temp}$ is computed independently from the input of the 3D attention module, since direct access to the 3D attention is not possible due to the adoption of Flash Attention (Dao et al., 2022). Specifically, For video latent $\mathbf{z} \in \mathbb{R}^{B \times (F \times H \times W) \times d}$ with batch size $B$, $F$ frames, spatial dimensions $H \times W$ and hidden size $d$, we reshape features by merging spatial dimensions into the batch size, yielding $\tilde{\mathbf{z}} \in \mathbb{R}^{(B \times H \times W) \times F \times d}$. Self-attention (Vaswani et al., 2023) is then applied along the frame axis:

$$A^{temp} = \text{Attn}(Q(\tilde{\mathbf{z}}), K(\tilde{\mathbf{z}})) \in \mathbb{R}^{(B \times H \times W) \times F \times F} \tag{5}$$

where $Q$ and $K$ denote the Query and Key heads, and $A^{temp}$ satisfies $\sum_{j=1}^{F} A^{temp}_{(b,i,j)} = 1$. The diagonal elements $A^{temp}_{ii}$ correspond to intra-frame attention, and the non-diagonal elements $A^{temp}_{ij}$ ($i \neq j$) represent cross-frame attention.

Next, the Cross-Frame Intensity (*CFI*) is calculated by averaging the non-diagonal elements of the attention map. The *CFI* is then multiplied by the enhance temperature parameter $\tau$ to enhance cross-frame correlations better:

$$CFI = \frac{1}{F(F-1)} \sum_{i=1}^{F} \sum_{\substack{j=1 \\ j \neq i}}^{F} A^{temp}_{ij}. \tag{6}$$

$$CFI_{enhanced} = \mathbf{clip}((\tau + F) \cdot CFI, 1). \tag{7}$$

Noticeably, the enhanced Cross-Frame Intensity (*CFI*_enhanced) is clipped at a minimum value of 1, which prevents severe weakening of cross-frame correlations during enhancement.

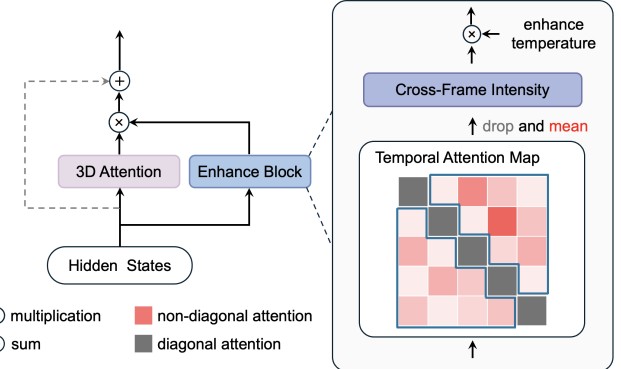

Figure 4: Overview of the Enhance Block. The block computes the average of non-diagonal elements from the temporal attention map as Cross-Frame Intensity (*CFI*). The *CFI* is scaled by the temperature parameter and fused back to enhance the temporal attention output.

Finally, the output of the Enhance Block (*CFI*_enhanced) is utilized to enhance the original temporal attention block output $\mathbf{O}_{attn}$ in the residual connection:

$$\mathbf{O}_{final} = CFI_{enhanced} \cdot \mathbf{O}_{attn} + \mathbf{H}. \tag{8}$$

where $\mathbf{H}$ represents the hidden states that are inputs of the attention block.

When $CFI_{enhanced}$ exceeds 1, indicating significant cross-frame information, the ratio of temporal attention block outputs is correspondingly amplified in $\mathbf{O}_{final}$. Otherwise, the connection defaults to a standard residual connection. Since $\mathbf{O}_{attn}$ is relatively small compared to $\mathbf{H}$, modest enhancements (small $CFI_{enhanced}$) to $\mathbf{O}_{attn}$ slightly affect the $\mathbf{O}_{final}$ distribution, enabling Enhance-A-Video to enhance cross-frame attention without substantially altering original temporal attention patterns. The complete analytical details are available in Appendix D.

The temporal attention difference map in Fig. 5 shows the difference between the temporal attention of the original CogVideoX model and w/ Enhance-A-Video, illustrating how Enhance-A-Video properly strengthens cross-frame attention without disrupting the initial temporal attention pattern. Specifically, certain non-diagonal elements (blue areas) are moderately increased (e.g., $0.9 \times 10^{-2}$), indicating enhanced cross-frame correlations. Meanwhile, the diagonal elements experience a minimal reduction ($3.3 \times 10^{-2}$ at most), which ensures stable intra-frame attention and preserves existing fine-grained visual details. More analysis can be found in Appendix C.

## 4 EXPERIMENTS

### 4.1 SETUP

To evaluate the effectiveness of our proposed Enhance-A-Video method, we conduct experiments on video generation models incorporating two types of attention mechanisms: 3D full attention and spatial-temporal attention. Specifically, we choose several representative models for each category:

**3D Full Attention Model:** Wan (Wan et al., 2025), HunyuanVideo (Kong et al., 2025), CogVideoX (Yang et al., 2024) and LTX-Video (HaCohen et al., 2024), which employ 3D full attention to model spatial and temporal dependencies simultaneously.

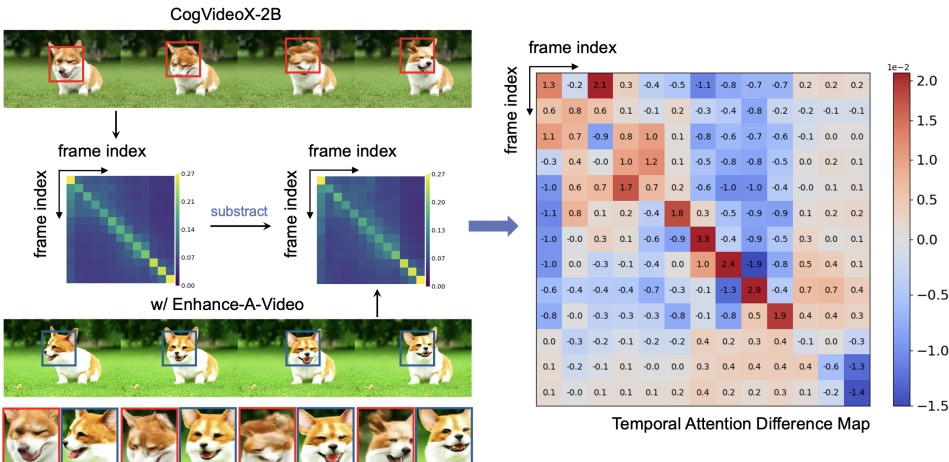

Figure 5: Temporal attention difference map between original CogVideoX model and w/ Enhance-A-Video of layer 29 at denoising step 50. Non-diagonal elements in the attention matrix of w/ Enhance-A-Video show higher values (shown in blue), while diagonal elements have reduced values (shown in red).

**Spatial-Temporal Attention Model:** Open-Sora (Zheng et al., 2024) and Open-Sora-Plan v1.0.0 (Lin et al., 2024), which decompose the attention mechanism into separate spatial and temporal components for computational efficiency and scalability.

Detailed experimental configurations are presented in Appendix E.1. We follow the original setup of these methods exactly and incorporate the Enhance Block exclusively into the temporal attention calculation of these models during the inference phase without additional retraining or fine-tuning. To facilitate the practical use of Enhance-A-Video, we provide the recommended range for the enhance temperature parameter in Appendix E.2.

## 4.2 3D FULL ATTENTION MODEL

frame index

(a)

Wan

w/ Enhance-A-Video

(b)

Wan

w/ Enhance-A-Video

Figure 6: Qualitative results of Enhance-A-Video on Wan. Captions: *(a) A cat **walks** on the grass, realistic. (b) A sheep and a cow.*

Wan (Wan et al., 2025) is a state-of-the-art text-to-video diffusion model recognized for producing high-resolution and temporally coherent videos from textual prompts. Fig. 6 demonstrates the

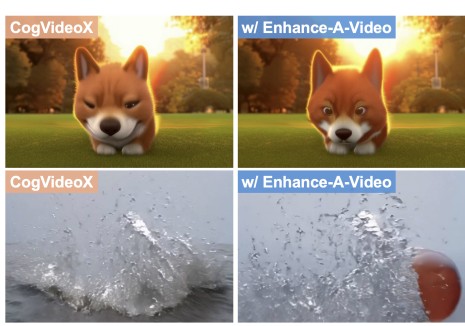

Figure 7: Qualitative results of Enhance-A-Video on more 3D full attention models. Videos generated by baselines (HunyuanVideo and CogVideoX) w/ Enhance-A-Video show significantly improved detail and quality compared to the original foundation models.

effectiveness of Enhance-A-Video on improving the foundation model with more dynamic and prompt-consistent video generation.

In the first case, Wan's output depicts a cat with unnatural fur lying motionless on the grass, whereas Enhance-A-Video generates a more realistically styled cat walking to the left, offering significantly better alignment with the prompt through improved motion dynamics and spatial content. In the second case, Enhance-A-Video generates the sheep and cow with enhanced texture sharpness, improved color fidelity, and clearer background details, resulting in frames that are more realistic and temporally consistent compared to those produced by the original model.

HunyuanVideo (Kong et al., 2025) is a large-scale text-to-video diffusion model that combines a causal 3D VAE, dual-to-single-stream Transformers, and a multimodal LLM encoder to generate high-quality videos. Our implementation of Enhance-A-Video augmentation in HunyuanVideo improves the model's video generation capabilities effectively. The results shown in Fig. 7 demonstrate that Enhance-A-Video consistently produces more realistic images with better details.

In the first row for HunyuanVideo, HunyuanVideo introduces conflicting artifacts, such as *duplicated right hands* and *unnatural head movements*. In contrast, Enhance-A-Video captures the baseball player's motion with greater fluidity and richer detail. In the second row, Enhance-A-Video enhances the appearance of the silver plane, making it more realistic.

By applying Enhance-A-Video to CogVideoX (Yang et al., 2024), we observe significant improvements in prompt-video consistency and visual detail. In the second row of Fig. 7 for CogVideoX, CogVideoX fails to accurately capture the prompt describing a "balloon full of water", generating only vague water splashes without the balloon. In contrast, the enhanced model produces videos that better align with the given prompts while delivering smoother transitions and clearer visuals. In addition, the results for LTX-Video, along with additional visual examples from the evaluated models, are provided in Appendix H.

### 4.3 SPATIAL-TEMPORAL ATTENTION MODEL

Open-Sora (Zheng et al., 2024) is an efficient text-to-video generation model that utilizes a decomposed spatial-temporal attention mechanism to balance computational efficiency and video quality. Incorporating the Enhance-A-Video augmentation into Open-Sora significantly improved temporal consistency and spatial detail preservation. As demonstrated in Fig. 8, the enhanced model produces videos with more natural motion transitions and more realistic visual details.

Open-Sora-Plan v1.0.0 (Lin et al., 2024) is a text-to-video generation model leveraging a multi-resolution latent diffusion framework for high-quality, temporally coherent videos. As shown in Fig. 8, Enhance-A-Video creates clearer leaves and sharper flower details, removing the blur seen in the baseline model. These improvements highlight Enhance-A-Video's ability to enhance cross-frame attention and produce visually high-quality videos.

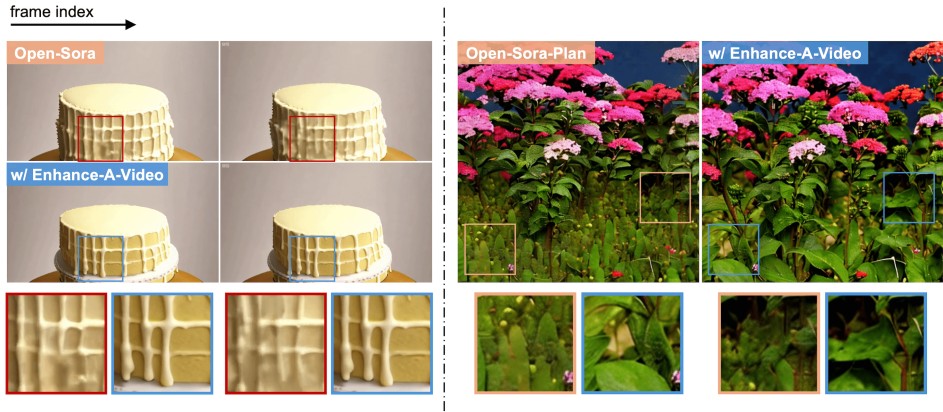

Figure 8: Qualitative results of Enhance-A-Video on spatial-temporal Attention models.

## 4.4 QUANTITATIVE ANALYSIS

Table 1: VBench results comparing baseline models and w/ Enhance-A-Video (EAV) across 5 evaluation criteria.

| Model | Subject Consistency ↑ | Temporal Flickering ↑ | Imaging Quality ↑ | Multiple Objects ↑ | Spatial Relationship ↑ |
|---|---|---|---|---|---|
| Wan 2.1 | 91.80 | 98.00 | 64.94 | 91.67 | 98.10 |
| w/ EAV | **92.18** | **98.44** | **65.18** | **92.78** | **99.59** |
| CogVideoX | 92.23 | 90.29 | 57.15 | 52.29 | 52.27 |
| w/ EAV | **92.93** | **91.77** | **58.56** | **54.42** | **56.82** |
| Open-Sora | 94.14 | 98.33 | 60.43 | 49.54 | 60.19 |
| w/ EAV | **94.27** | **98.92** | **61.04** | **53.92** | **64.58** |

We employ VBench (Huang et al., 2024) for automatic evaluation across multiple video metrics. CogVideoX and Open-Sora use the standard VBench prompt set, while Wan is evaluated on 100 randomly sampled prompts due to computational constraints.

The results in Tab. 1 demonstrate that Enhance-A-Video consistently strengthens video generation across multiple dimensions: it improves subject consistency and temporal stability by amplifying cross-frame attention adaptively, leading to smoother and more coherent motion, while also boosting imaging quality through better preservation of fine details. Moreover, Enhance-A-Video enhances complex scene understanding and generation, as reflected in higher scores for multiple objects and spatial relationships, indicating more accurate object interactions and structural alignment.

Importantly, these benefits appear across different backbone models (Wan 2.1, CogVideoX, Open-Sora), highlighting our method's robustness and generalizability as a lightweight, training-free plug-in that enhances temporal coherence and perceptual fidelity without retraining costs. Further results are presented in Appendix E.4.

Table 2: User study results comparing baseline models and w/ Enhance-A-Video across evaluation criteria. We compute the proportion of votes received by the baseline and our method respectively.

| Model | Overall | Temporal Consistency | Prompt-Video Consistency | Visual Quality |
|---|---|---|---|---|
| Baseline | 20.30 | 22.73 | 21.82 | 16.36 |
| w/ Enhance-A-Video | **79.70** | **77.27** | **78.18** | **83.64** |

Furthermore, we evaluated video quality through a blind user study of 110 participants. Each person compared two videos generated from the same text prompt and random seed - one from baseline models and one from w/ Enhance-A-Video. The videos were shown in random order to prevent bias. Participants chose which video they preferred based on three criteria: temporal consistency, prompt-video consistency, and overall visual quality. The detailed setting is given in Appendix E.3.

Tab. 2 presents the main user study results for chosen models and w/ Enhance-A-Video of each evaluation criterion. The results show that models using Enhance-A-Video received the majority of preference, demonstrating that Enhance-A-Video notably enhances the text-to-video models' performance in all evaluated aspects:

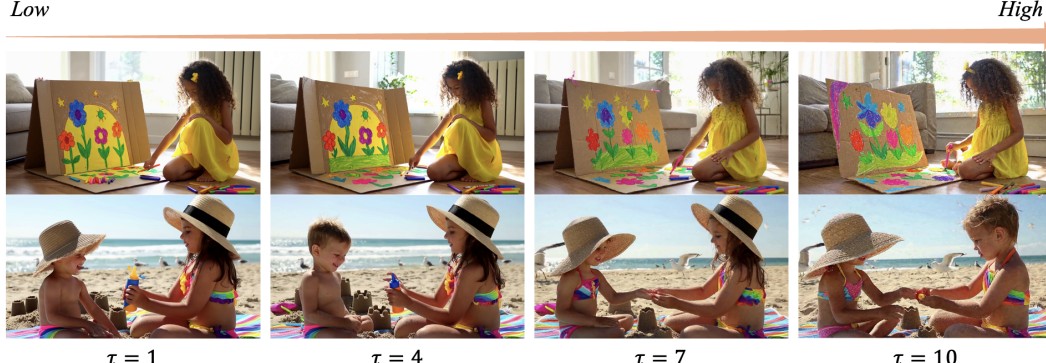

Figure 9: Ablation study on the enhance temperature parameter in the Enhance Block. Moderate values balance temporal consistency and visual diversity, while extreme values degrade performance.

**Temporal Consistency.** The usage of Cross-Frame Intensity (*CFI*) and the enhance temperature parameter strengthens cross-frame connections. This results in smoother motion transitions and improved frame-to-frame alignment, which creates a more stable and coherent visual experience in the generated video.

**Prompt-Video Consistency.** In diffusion-based video generation, video frames are progressively denoised based on the prompt. However, the lack of temporal attention in cross-frame information transmission causes the semantic alignment between the video and the prompt to deviate gradually during generation. Enhancing cross-frame information by Enhance-A-Video ensures that objects and actions in the scene remain consistent with the prompt. This smooth semantic evolution avoids abrupt or inconsistent content, improving the alignment between the generated video and the given prompt.

**Visual Quality.** By using *CFI* and the enhanced temperature parameter, the model makes better use of information from contextual frames to improve details, especially in object textures and edges. The improved cross-frame attention smooths the denoising process and reduces random changes, allowing the model to generate more consistent motion and avoid unrealistic movements.

### 4.5 ABLATION STUDY

**Impact of Temperature.** To better understand the impact of the temperature parameter, we conduct an ablation study by varying the enhance temperature parameter in the Enhance Block. Results in Fig. 9 indicate that moderate temperature values achieve the best balance between temporal consistency and diversity, while extreme values (too low or too high) will degrade performance.

**Minimal Overhead.** To evaluate the inference efficiency of the proposed Enhance-A-Video (EAV) method, we conducted an ablation study on two prevailing video generation models in Tab. 3 using 1 A100 GPU. These negligible increases in the two models indicate that the Enhance-A-Video method is highly efficient and scales well when integrated into large video generation models. Additional ablation studies are provided in Appendix F.

Table 3: Comparison of inference efficiency for HunyuanVideo and CogVideoX models with and without Enhance-A-Video.

| Model | Time (min) | | Overhead |
| --- | --- | --- | --- |
| | w/o EAV | w/ EAV | |
| HunyuanVideo | 50.32 | 50.72 | **0.8%** |
| CogVideoX | 1.53 | 1.57 | **2.1%** |

## 5 CONCLUSION

This paper presents Enhance-A-Video, a simple yet effective method that improves temporal consistency and visual quality in DiT-based video generation. By pioneering the exploration of cross-frame information and the temperature concept in DiT blocks, the method offers a straightforward yet powerful solution for video generation enhancement. Its robust generalization and ease of implementation suggest promising future developments in better video generation.

## REPRODUCIBILITY STATEMENT

We employ open-source models and prompts, and provide the code necessary to reproduce our results in the supplementary material. After the blind review period, we will release the complete codebase. Detailed configurations of the evaluated models and the recommended range for the enhance temperature parameter are presented in Appendix E.1 and E.2.

## ETHICS STATEMENT

After reviewing the conference's ethical guidelines, we believe this work poses no foreseeable ethical concerns. The proposed algorithms focus on general video generation and do not involve human subjects, harmful insights, conflicts of interest, privacy or security issues, legal compliance, or research integrity concerns.

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

## A  USAGE OF LARGE LANGUAGE MODELS (LLMS)

In this paper, the ideas, experiments, and manuscript writing were all carried out by the authors. LLMs were only used for minor language polishing, without contributing to the conception, experimental design, or substantive content of the work.

## B  LIMITATIONS AND FUTURE WORK

In our approach, the enhancement temperature parameter needs to be manually tuned for each DiT-based text-to-video model. In future work, we plan to develop an adaptive temperature mechanism using RLHF (Christiano et al., 2017; Li et al., 2024; 2025a) to adjust this parameter based on the specific prompt context automatically. Besides, we focused solely on enhancing temporal attention without addressing spatial attention or cross-attention mechanisms, which are crucial for preserving spatial coherence and prompt alignment. Future work could explore incorporating these mechanisms to improve spatial video quality and semantic consistency.

## C  TEMPERATURE METHOD COMPARISON

In Fig. 10(a) and (b), where the temperature parameter $\tau$ and Cross-Frame Intensity are directly applied in temporal attention calculation separately as presented in Equation 9 and 10, the diagonal elements (e.g., 27.4, 6.3) show a significant weakening of intra-frame attention, leading to the severe loss of spatial details and resulting in blurry and unrealistic textures. Additionally, the large negative values in the off-diagonal regions indicate an overabundant distributed enhancement of cross-frame attention, resulting in limited improvement in video quality.

$$\text{Attn}(Q, K, V) = \text{softmax}\left(\frac{QK^{\top}}{\tau \cdot \sqrt{d_k}}\right) V \tag{9}$$

$$\text{Attn}(Q, K, V) = \text{softmax}\left(\frac{QK^{\top}}{\textit{\textbf{CFI}}_{enhanced} \cdot \sqrt{d_k}}\right) V \tag{10}$$

In contrast, Fig. 10(c) using the Enhance-A-Video method shows modest changes along the diagonal, with values close to zero, preserving intra-frame attention and maintaining fine-grained details. Moreover, the negative values in the off-diagonal regions (e.g., -1.3, -0.9) reflect a targeted and moderate enhancement of cross-frame attention, significantly improving motion coherence and overall video quality.

## D  CFI DISTRIBUTION AND L2 NORM PROPORTION IN RESIDUAL CONNECTION

The $CFI_{enhanced}$ values in Fig. 11(a) range between $1.12 - 1.18$, indicating a modest enhancement of keyframes containing important temporal information. Fig. 11(b) shows two low proportions calculated as follows:

$$\text{prop}_{\text{CogvideoX}} = \frac{||\mathbf{O}_{\text{attn}}||_2}{||\mathbf{H}||_2} \tag{11}$$

$$\text{prop}_{\text{w/ Enhance-A-Video}} = \frac{||CFI_{enhanced} \cdot \mathbf{O}_{\text{attn}}||_2}{||\mathbf{H}||_2} \tag{12}$$

suggesting that attention outputs are relatively small compared to hidden states in the residual connection. Consequently, applying $CFI_{enhanced}$ to attention outputs rather than attention allows for enhancing important information with minimal disruption to the original attention distribution. Thus, Enhance-A-Video improves temporal consistency while preserving existing spatial details.

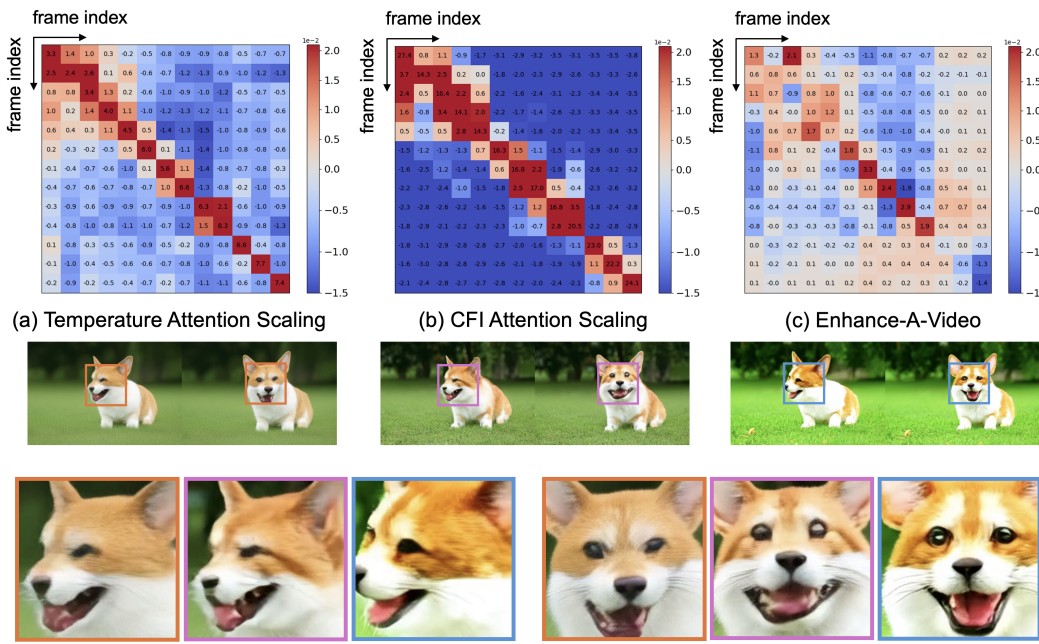

Figure 10: Temporal attention difference maps and corresponding generated videos comparing three temperature enhancement methods. (a) Temperature Attention Scaling $\tau = 1.1$. (b) CFI Attention Scaling. (c) Enhance-A-Video Method.

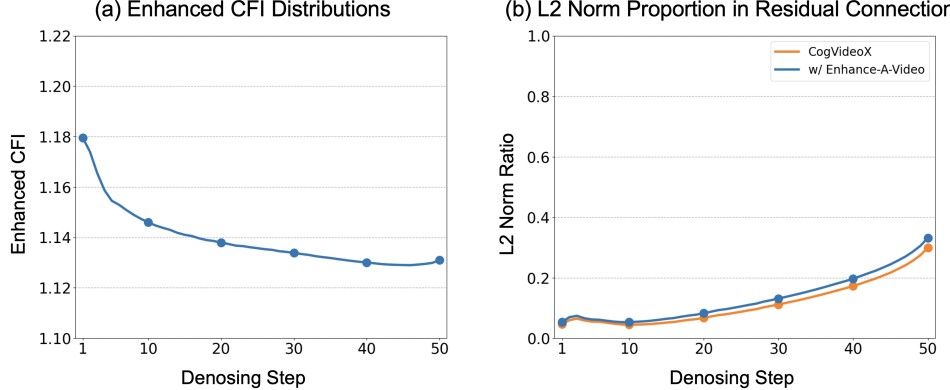

Figure 11: (a) The distribution of $CFI_{enhanced}$ during the inference of CogVideoX w/ Enhance-A-Video in layer 4. (b) The proportion of $l_2$ norms between $\mathbf{O}_{attn}$ and $\mathbf{H}$ in residual connection in layer 4.

# E   MORE EXPERIMENTAL INFORMATION

## E.1   EVALUATED MODELS AND SETTINGS

We evaluated Enhance-A-Video using widely adopted open-source models with default baseline parameters. Detailed settings are given below, and all experiments were conducted on NVIDIA H100 or A100-80GB GPUs.

## E.2   IDEAL RANGE FOR ENHANCE TEMPERATURE PARAMETER

To guide users in selecting an appropriate configuration, we provide recommended ranges for the enhance temperature parameter $\tau$ tailored to each foundation model. These ranges are determined

Table 4: Specific settings of evaluated models.

| Model | Size | Video Resolution | Duration | Frame Rate (FPS) |
|---|---|---|---|---|
| Wan2.1 | 14B | 480p | 5s | 16 |
| HunyuanVideo | 13B | 720p | 5s | 24 |
| CogVideoX | 5B | 480p | 6s | 8 |
| LTX-Video | 1.9B | 512p | 5s | 24 |
| Open-Sora 1.2 | 1.1B | 480p | 4s | 16 |
| Open-Sora-Plan v1.1.0 | 1.2B | 512p | 2s | 30 |

based on empirical observations that balance generation quality and consistency across diverse prompts. By offering model-specific $\tau$ intervals in Tab. 5, we facilitate more efficient tuning and ensure that users can achieve optimal results without exhaustive manual search.

Table 5: Recommended $\tau$ ranges for each foundation model.

| Model | Range |
|---|---|
| HunyuanVideo | **3–5** |
| CogVideoX-2B | **1–3** |
| LTX-Video | **5–7** |
| Open-Sora | **1–3** |

### E.3 DETAILED SETTINGS FOR QUANTITATIVE ANALYSIS

To evaluate the effectiveness of Enhance-A-Video on different foundation models in practical scenarios, we conducted a user study based on 15 prompts sampled from the VBench benchmark (Huang et al., 2024). The number of evaluated samples per model is summarized in Tab. 6. We selected HunyuanVideo as the primary model for our quantitative analysis to ensure a more representative evaluation of Enhance-A-Video's effectiveness on a high-performing open-source baseline.

Table 6: Number of evaluated samples per model in the user study.

| Model | Number of Samples |
|---|---|
| HunyuanVideo | **9** |
| CogVideoX | **2** |
| LTX-Video | **2** |
| Open-Sora | **2** |

### E.4 COMPARISON WITH UNET-BASED GENERATION ENHANCEMENT METHODS

We selected two representative UNet-based training-free enhancement methods for comparison with our approach. Since Flash Attention prevents direct access and modification of temporal attention in existing 3D attention models, our comparison primarily focused on Open-Sora, which employs spatial–temporal attention.

#### E.4.1 COMPARISON WITH BYTHEWAY

As shown in 7, existing UNet-based methods fail to effectively enhance video generation due to structural differences in UNet and DiT models, whereas our first DiT-oriented approach resolves this limitation without incurring additional cost.

Table 7: VBench results comparing baseline models, w/ ByTheWay and w/ Enhance-A-Video (EAV) across 5 evaluation criteria.

| Model | Subject Consistency ↑ | Temporal Flickering ↑ | Imaging Quality ↑ | Multiple Objects ↑ | Spatial Relationship ↑ |
|---|---|---|---|---|---|
| Open-Sora | 94.14 | 98.33 | 60.43 | 49.54 | 60.19 |
| w/ ByTheWay | 94.19 | 98.33 | 60.41 | 47.72 | 60.24 |
| w/ EAV | **94.27** | **98.92** | **61.04** | **53.92** | **64.58** |

### E.4.2 COMPARISON WITH FREEU

We experimented to assess the performance of FreeU on the Open-Sora framework using its default settings ($b_1 = 1.1$, $b_2 = 1.1$). As illustrated in Fig. 12, applying FreeU only to the last two layers results in overly smooth videos with noticeable loss of detail, while extending it to more layers (e.g., four) causes the output to lose nearly all meaningful content. These results suggest that FreeU is not well-suited for DiT-based video generation, as it was originally developed for UNet-based image generation and does not effectively capture temporal dependencies or adapt to architectural differences. In contrast, Enhance-A-Video is specifically designed for DiT-based video models, efficiently addressing temporal attention while maintaining inference overhead comparable to FreeU's usage in image generation.

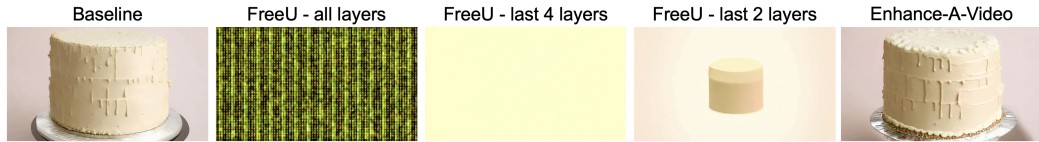

Figure 12: Qualitative results of FreeU on Open-Sora. FreeU fails to preserve structure when applied to DiT, especially with more layers. Enhance-A-Video improves visual detail and temporal consistency.

## F MORE ABLATION STUDY

### F.1 ABLATION STUDY ON CLASSIFIER-FREE GUIDANCE

In our experiments, we adopted the default classifier-free guidance (CFG) values specified for each foundation model, as summarized in Tab. 8:

Table 8: Default classifier-free guidance (CFG) settings for each foundation model.

| Model | Default CFG |
|---|---|
| HunyuanVideo | **6.0** |
| CogVideoX | **6.0** |
| LTX-Video | **3.0** |
| Open-Sora | **7.0** |

To further investigate the influence of CFG on the effectiveness of our method, we conducted ablation studies using CogVideoX by generating a 6-second, 480p video with the prompt "A cute happy Corgi playing in park". We varied the CFG values across 3.0, 5.0, 7.0, and 9.0, applying Enhance-A-Video in each case.

As shown in Fig. 13, the impact of Enhance-A-Video is highly dependent on the base video quality determined by the CFG setting. At higher CFG values (7.0 and 9.0), where the base outputs are of relatively high quality, Enhance-A-Video consistently improves spatial fidelity without introducing artifacts. However, at lower CFG values (3.0 and 5.0), where the base generation quality is poor, Enhance-A-Video is limited in its ability to recover detail and consistency, as it is designed to enhance—not replace—the underlying generative capacity of the model.

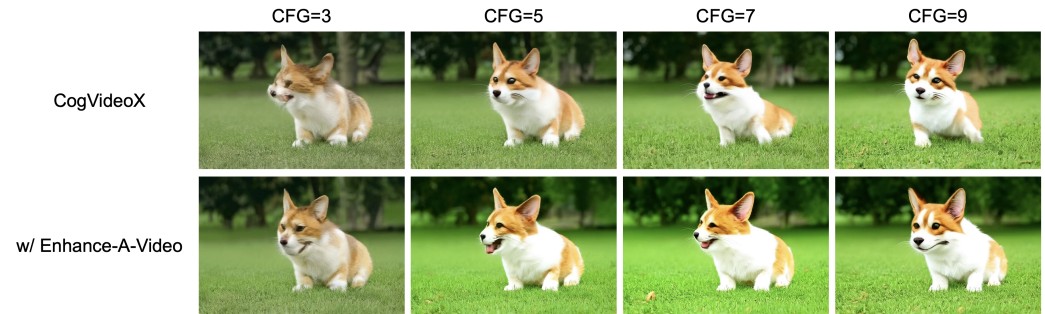

Figure 13: Qualitative results of Enhance-A-Video with different CFG settings on CogVideoX. Caption: *A cute and happy Corgi playing in the park.*

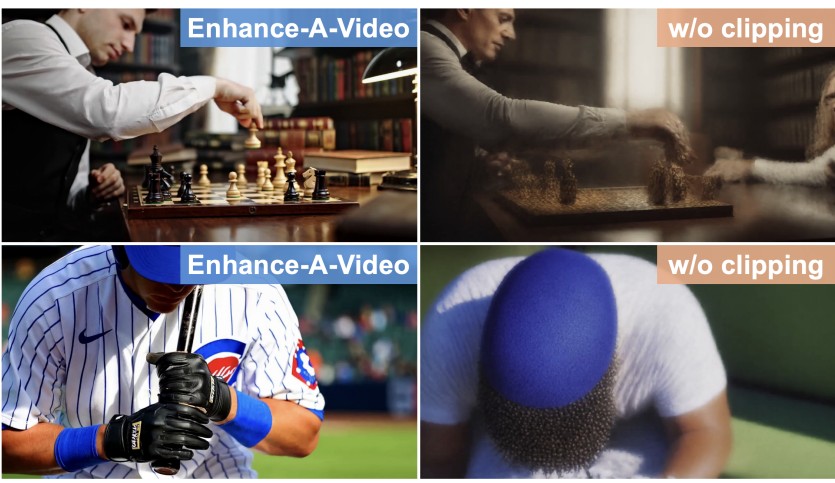

Figure 14: Visual comparison of video generation results with and without the clipping mechanism in the Enhance Block.

## F.2 EFFECTS OF CLIPPING.

Fig. 14 illustrates that applying the clipping effectively stabilizes cross-frame attention, resulting in clearer visuals and smoother motion. Without clipping, the model produces noticeable artifacts such as motion blur and distorted details, highlighting the necessity of clipping for maintaining temporal consistency and preserving spatial fidelity.

## F.3 DISCUSSION ON LONG VIDEO GENERATION

We employed HunyuanVideo to produce a 9-second video comprising 217 frames—well beyond its default limit of 129 frames—and assessed the performance of Enhance-A-Video on the video.

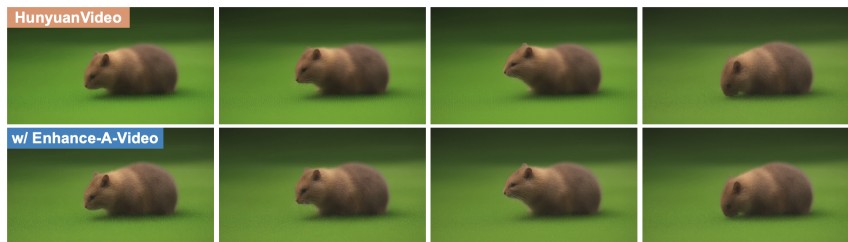

Figure 15: Visual comparison of video generation results on long video generation.

The video generated by HunyuanVideo in Fig. 15 shows poor visual quality, with blurred spatial details and low prompt consistency (e.g., the output resembles a mouse rather than a cat). This degradation stems from the model's limited capacity to handle longer videos, likely due to the insufficient availability of high-quality long video data during training. Addressing this issue would require improvements in training data, model architecture, and overall scale.

Since Enhance-A-Video operates solely during the inference stage in a training-free way, its effectiveness depends heavily on the original video quality. While it can enhance decent videos with more details or better temporal consistency, it struggles when the input is of very low quality - as in this case - because the core limitations lie in the training process, not inference.

## G  CAPTIONS FOR FIGURE 1

Caption 1 (top row): A young girl with curly hair, wearing a bright yellow dress, sits cross-legged on a wooden floor, surrounded by an array of colorful markers and crayons. She carefully colors a large piece of cardboard, her face a picture of concentration and creativity. The cardboard, propped up against a cozy living room couch, is filled with whimsical drawings of flowers, stars, and animals. Sunlight streams through a nearby window, casting a warm glow over her workspace. Her small hands move deftly, adding vibrant hues to her imaginative artwork, while her expression reflects pure joy and artistic focus.

Caption 2 (bottom row): A young girl, wearing a wide-brimmed straw hat and a colorful swimsuit, carefully applies sunblock to her younger brother's face on a sunlit beach. The boy, with **sandy hair** and a playful grin, sits patiently on a striped beach towel, surrounded by **sandcastles** and **beach toys**. The gentle waves of the ocean provide a soothing soundtrack as seagulls call in the distance. The girl's hands move with care, ensuring every inch of his face is protected, while the sun casts a warm glow over the scene, highlighting the siblings' bond and the carefree joy of a summer day by the sea.

## H  MORE QUALITATIVE RESULTS

LTX-Video (HaCohen et al., 2024) is a real-time latent text-to-video diffusion model that generates high-quality, temporally consistent videos efficiently. The integration of Enhance-A-Video into LTX-Video further improves temporal consistency and enhances spatial details. As exhibited in Fig. 16, the enhanced model produces videos with sharper textures, more vivid colors, and smoother transitions compared to the baseline LTX-Video.

The snow-covered mountains (top row) and river scene (bottom row) generated by Enhance-A-Video display clearer structures and more natural color gradients, while the baseline results appear less detailed and slightly blurred. This demonstrates that Enhance-A-Video effectively strengthens cross-frame attention, leading to more realistic and visually appealing videos.

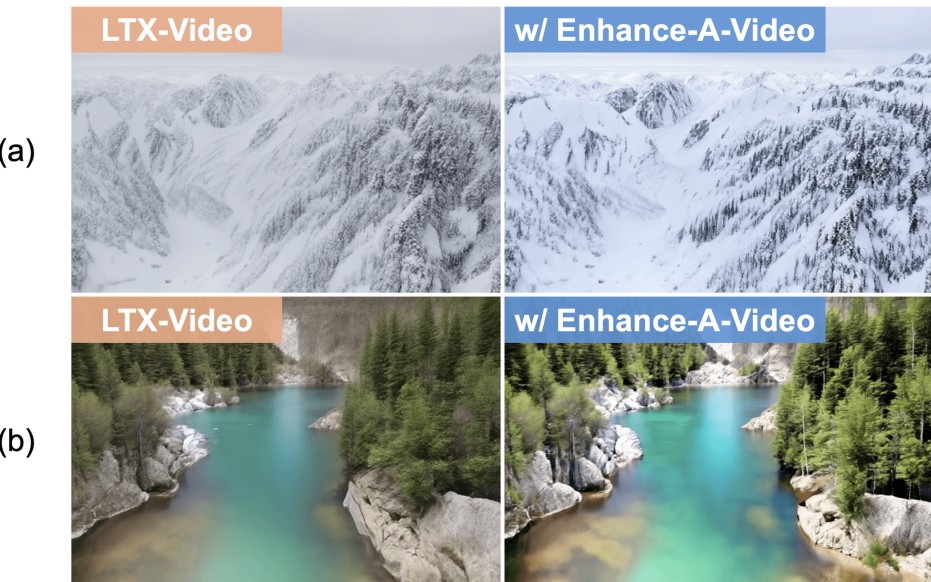

Figure 16: Qualitative results of Enhance-A-Video on LTX-Video. Captions: *(a) The camera pans over snow-covered mountains, revealing jagged peaks and deep, narrow valleys. (b) An emerald-green river winds through a rocky canyon, forming reflective pools amid pine trees and brown-gray rocks.*

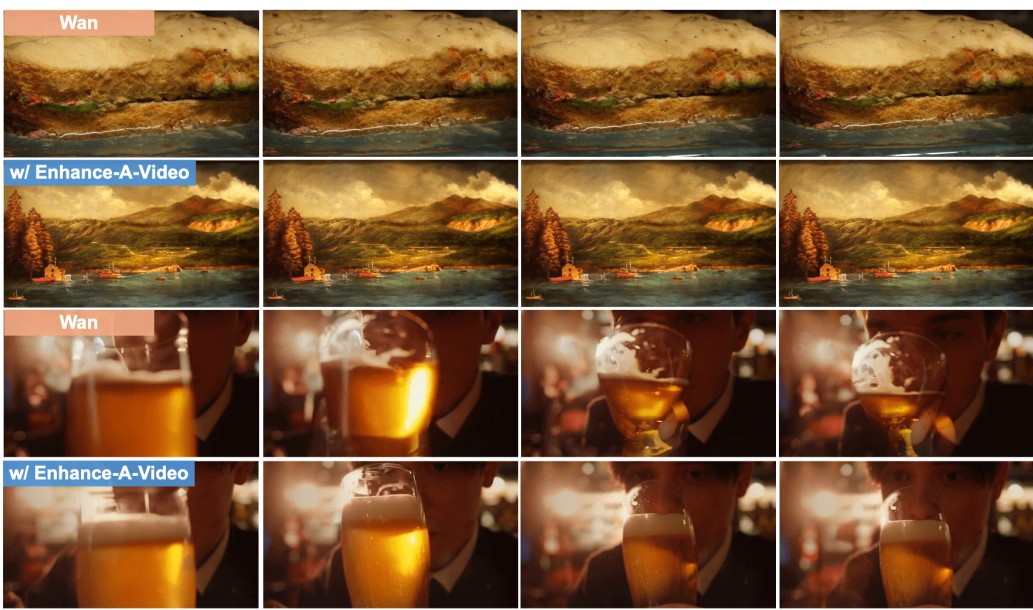

Figure 17: More qualitative results of Enhance-A-Video on Wan. Captions: *(a) A tranquil tableau of bar. (b) A person is tasting beer.* The integration of Enhance-A-Video facilitates prompt-consistent video generation while ensuring smoother and more natural transitions.

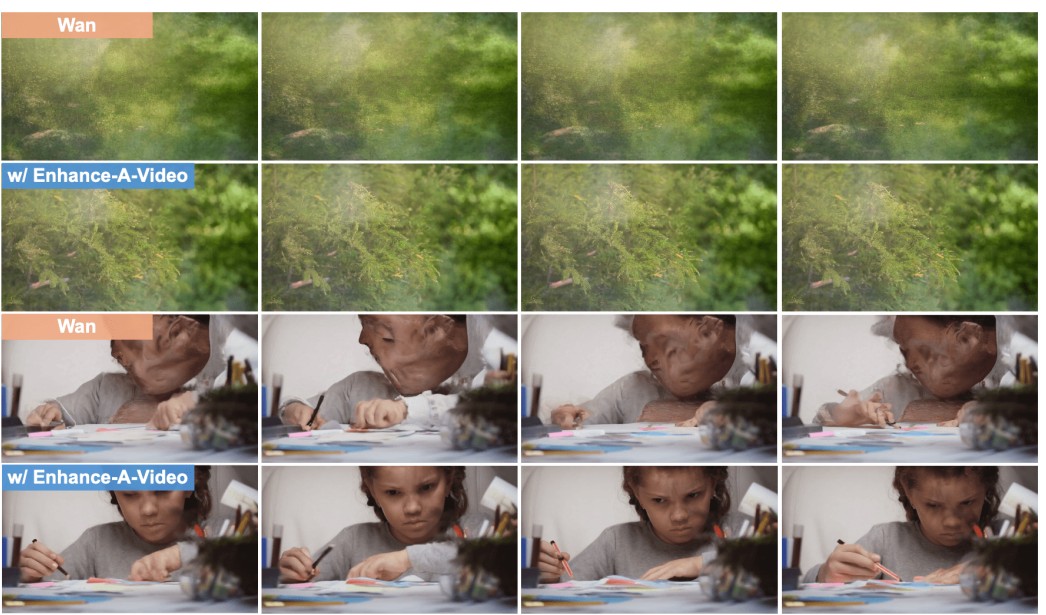

Figure 18: More qualitative results of Enhance-A-Video on Wan. Captions: *(a) A tranquil tableau of alley. (b) A person is drawing.* The use of Enhance-A-Video enhances image quality, allowing the model to more effectively leverage contextual information.

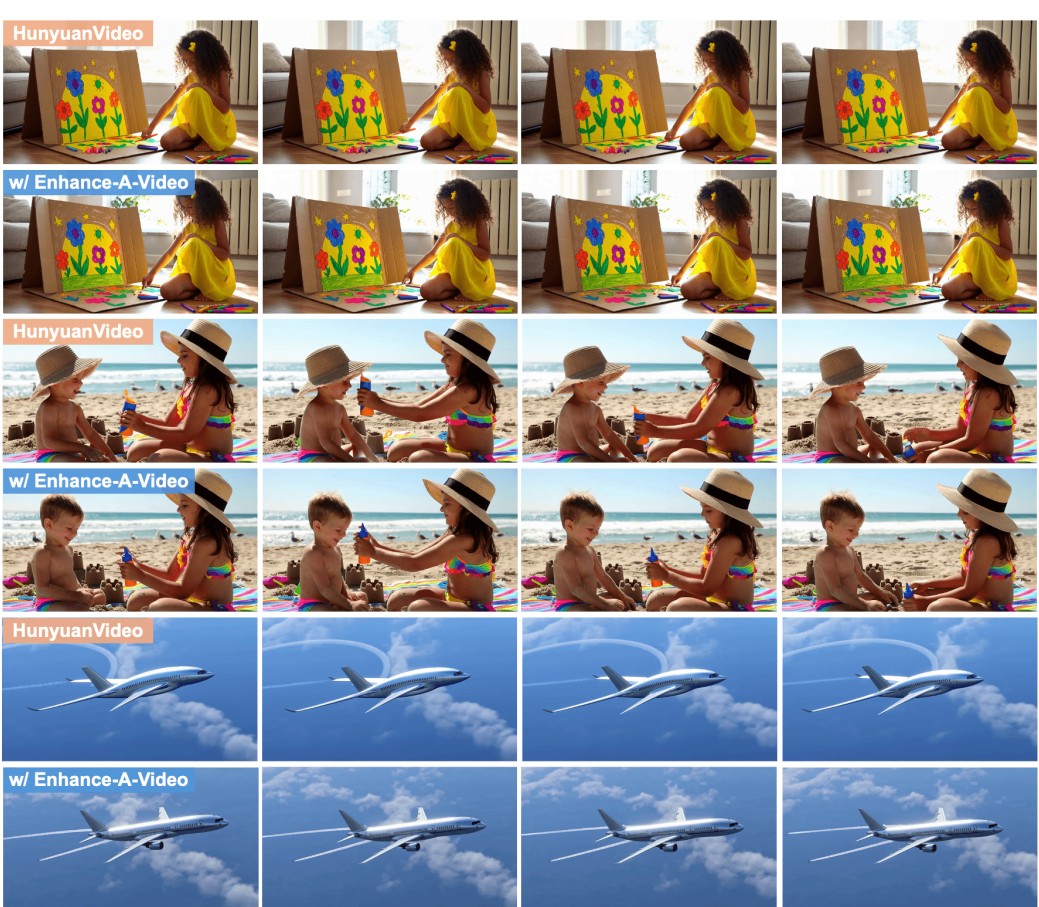

Figure 19: More qualitative results of Enhance-A-Video on HuanyuanVideo. The application of Enhance-A-Video enriches visual details and ensures prompt-consistent video generation, resulting in more realistic outputs.

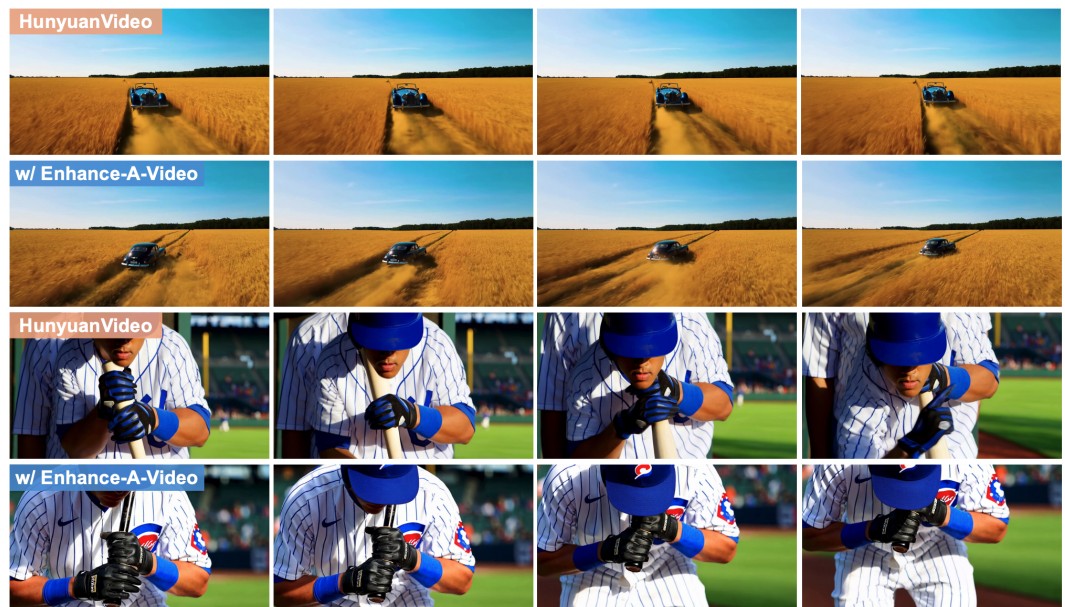

Figure 20: More qualitative results of Enhance-A-Video on HunyuanVideo. Captions: *(a) An antique car drives along a dirt road through golden wheat fields. Dust rises softly as wheat brushes against the car with distant trees meeting a blue sky. (b) A baseball player grips a bat in black gloves, wearing a blue-and-white uniform and cap, with a blurred crowd and green field highlighting his focused stance.* Enhance-A-Video consistently generates more realistic frames with finer details and more natural motion.

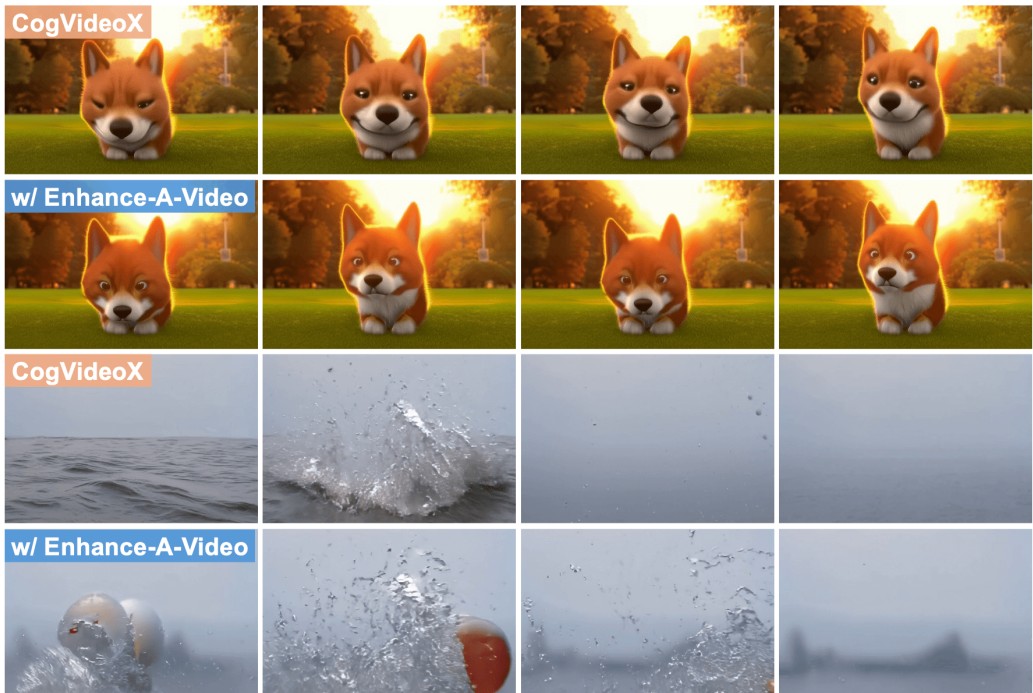

Figure 21: More qualitative results of Enhance-A-Video on CogVideoX. Captions: *(a) A cute and happy Corgi playing in the park, in a surrealistic style. (b) Balloon full of water exploding in extreme slow motion.* The enhanced model generates videos that align more closely with the given prompts, while providing smoother transitions and sharper visuals.

