# OpenReview forum: "Enhance-A-Video: Better Generated Video for Free"
_ICLR.cc/2026/Conference — ICLR 2026 Conference Withdrawn Submission_

### Official Review · Reviewer_kFrD · 2025-10-27

**Soundness:** 2
**Presentation:** 2
**Contribution:** 2
**Rating:** 2
**Confidence:** 4

**Summary:**

This paper introduces Enhance-A-Video, a training-free and plug-and-play method designed to improve the quality of videos generated by pre-trained DiT models. The authors identify that existing DiT models often suffer from temporal inconsistencies and poor visual details because their temporal attention mechanisms focus excessively on intra-frame information rather than cross-frame correlations. The proposed Enhance Block is a simple module that calculates the Cross-Frame Intensity (CFI) from the temporal attention map and uses it to adaptively scale the attention block's output within the residual connection. This approach aims to enhance temporal consistency and visual fidelity without requiring any model retraining or fine-tuning.

**Strengths:**

1. The method is training-free and functions as a plug-and-play module. This allows it to be integrated into various existing, pre-trained DiT-based video generation models without any costly retraining or fine-tuning.

2. The Enhance Block introduces negligible computational overhead during inference.

**Weaknesses:**

1. Conceptual Contradiction of Temperature

A primary weakness is the conceptual contradiction in the definition and application of temperature (tau).

-  In Section 3.2, the paper introduces 'temperature' in its classical sense: a parameter applied inside the softmax function to modulate the probabilistic distribution of attention weights without altering the feature scale (Eq. (4))
-  However, the proposed enhance temperature in Section 3.3 is used in a completely different mechanism. As shown in Eq. (7), it is summed with the frame count F and used to scale the CFI value, an operation external to the softmax function.

Although the authors intentionally dismiss the classical method (Eq. (4)) as a naive strategy, their co-opting of the same term for two mechanically disparate operations creates significant confusion and undermines the paper's methodological clarity.

2. Lack of Theoretical Justification for Heuristic Formula

The core of the proposed method, Eq. (7), appears to be an empirical heuristic lacking clear theoretical justification.

- The operation of summing the temperature parameter tau with the number of frames F is not derived from any stated principle.
- The paper provides no deep analysis or ablation to justify why this specific formulation is an optimal or principled way to enhance cross-frame correlations.
- This suggests the method may be an engineering trick discovered to work in a specific experimental setup, rather than a generalizable principle.

3. Risks of Altering Feature Scales in a Training-Free Method

The most significant potential flaw is the method's direct alteration of intermediate feature scales within a training-free framework.

- Eq. (8) explicitly modifies the norm of the attention output by multiplying O_attn with the scalar CFI_enhanced.
- Pre-trained generative models like DiT are highly sensitive to the statistical distributions of the features they were trained on. Introducing features with an altered, out-of-distribution scale risks destabilizing subsequent layers and the entire denoising process, which could lead to severe artifacts.
- The authors' defense for this in Appendix D (Fig. 11(b))  is that the norm of the attention output is relatively small compared to the norm of the hidden state.
- This defense is tenuous. It relies on a single empirical observation on the CogVideoX model and does not guarantee that this property holds universally across all DiT architectures. The method's stability precariously relies on the assumption that O__attn's contribution to the residual stream is always negligible, which may not be a safe or generalizable assumption.

**Questions:**

- Beyond the high-level effect of smoothing the distribution, the proposed enhance temperature shares no mechanistic commonality with the classical temperature parameter defined in Section 3.2. Regarding the conceptual contradiction of temperature, could the authors elaborate on the theoretical link, if any, between the classical softmax modulator defined in Section 3.2 (Eq. 4) and the mechanically disparate scaling factor in Section 3.3 (Eq. 7), which is summed with the frame count F?
- Given this terminological confusion, would introducing a distinct term, such as enhancement strength, not have been clearer for the proposed mechanism in Eq. (7)?
- The core formula in Eq. (7) appears to be an empirical heuristic. What is the theoretical justification for summing the tau parameter with the frame count F?
- Could the authors detail the derivation process for Eq. (7)?
- How can the method's direct alteration of intermediate feature scales (Eq. 8) be considered stable within a training-free framework, given that it introduces out-of-distribution feature scales to a pre-trained model?

---

### Official Review · Reviewer_Fgea · 2025-10-31

**Soundness:** 3
**Presentation:** 3
**Contribution:** 2
**Rating:** 4
**Confidence:** 4

**Summary:**

The paper proposes Enhance-A-Video, which is a training-free and plug-and-play algorithm for DiT-based text-to-video models. The key idea is to compute a Cross-Frame Intensity (CFI) from the non-diagonal entries of the temporal attention map, adjust it with a manually tuned temperature, and inject the result into the residual connection of the temporal attention block. The method is tested with several DiT-based video generation models (Wan2.1, HunyuanVideo, CogVideoX, LTX-Video, Open-Sora) and demonstrates superior performance on VBench against the baselines with negligible overhead.

**Strengths:**

1. The proposed Enhance Block is conceptually clean, allowing it to be easily integrated into temporal attention modules in existing DiT-based video generation models without re-training or fine-tuning.
2. The proposed method improves the performance of the existing models with <3% runtime increase on A100.
3. Enhance-A-Video demonstrates its superiority across 3D-full and spatial-temporal attention video generation models (Wan2.1, HunyuanVideo, CogVideoX, LTX-Video, Open-Sora), suggesting decent robustness and broad applicability.

**Weaknesses:**

1. The 110-participant preference study is promising, yet drawn from only 15 prompts with an uneven per-model sample count. More prompts and counterbalancing would strengthen claims.
2. Why can’t model training learn the calibrated attention pattern? Your observation is that learned temporal attention concentrates on diagonals, under-exploiting cross-frame cues. Couldn’t one add a regularizer encouraging non-diagonals so that the model learns similar CFI-like balancing during training?
3. The temperature must be manually tuned per model. Will this value also depend on different seeds or even prompts?
4. The paper does not provide any video examples to support its claims of improved temporal consistency. This omission makes it difficult to judge whether the method truly enhances motion coherence or inadvertently over-constrains frame-to-frame dynamics. Intuitively, amplifying cross-frame attention might suppress desirable temporal variation/dynamics.
5. The paper claims that the proposed method improves many aspects of video generation and fixes various artifacts, but such broad benefits seem unjustified and overclaimed. The technique is simple, and it is unclear why it could consistently enhance so many dimensions. The shown examples are few; more systematic experiments and diverse cases are needed to support the claims.

**Questions:**

How does the proposed method work on other types of video generation models? e.g., image-to-video models, sound-to-video models, etc.

---

### Official Review · Reviewer_bnzY · 2025-10-31

**Soundness:** 3
**Presentation:** 2
**Contribution:** 2
**Rating:** 2
**Confidence:** 4

**Summary:**

This work presents Enhance-A-Viseo, it is a training-free, plug-and-play enhancement for DiT-based video generation. By softly amplifying cross-frame attention via a residual “cross-frame intensity” and temperature scaling, it boosts temporal coherence and visual fidelity without extra parameters or noticeable overhead, yielding consistent improvements across multiple leading DiT-based video generation models.

**Strengths:**

1. This paper is well-structured and easy to follow, making it accessible for readers at different levels of expertise.
2. The proposed Enhance-A-Video is a training-free and plug-and-play method, it is easy to integrate it with existing DiT-based T2V models, including SOTA models like WAN and HunyuanVideo.
3. The authors have conducted extensive experiments to demonstrate the effectiveness of Enhance-A-Video.

**Weaknesses:**

1. **The visual improvement is not significant**. The visual comparison depicted in Fig. 1, Fig. 6 (b) and Fig. 8 (left) does not present a significant performance gain compared to the original results. Moreover, I doubt that the results in Fig. 1 is actually a cherry-picked one, since current T2V models possess limited capability (or domain knowledge) in generating coherent limb features, but the authors claimed that their method can tackle these issues in a training-free manner without modifying model weights. Similar concerns also come from Fig. 18; I believe the original performance of the Wan2.1-14B model would not produce the blurry and distorted baseline shown by the authors, suggesting this baseline may have been carefully selected to exaggerate the improvement.
2. **The relationship between the proposed temporal attention operations and the improvement in key visual details is not well clarified**. As far as I concerned, such operations mainly contribute to temporal coherency. How can Enhance-A-Video facilitate the synthesis of image details like the human hand and the dog face? The authors can elaborate on the explanation of such phenomena.
3. **The paper’s presentation of results is inappropriate**. Most generated videos comprising dozens of frames are represented by only one to four frames, which is insufficient to judge the method’s true efficacy. Because the approach focuses on modifying temporal attention, potential impairments to long-range consistency cannot be ruled out; releasing longer frame sequences is therefore essential for a fair assessment.

**Questions:**

Please see weaknesses.

---

### Official Review · Reviewer_gH7t · 2025-11-04

**Soundness:** 3
**Presentation:** 3
**Contribution:** 3
**Rating:** 8
**Confidence:** 5

**Summary:**

the paper proposes a training-free framework for improving visual quality and prompt coherence of the video generation model

**Strengths:**

- the paper presents convincing visualizations and clearly justifies its motivation. the idea of using non-diagonal temporal attention with temperature for better temporal consistency seems to novel for video generation tasks. the module design is simple and intuitive
- the paper conducted a user study to validate the effectiveness of the proposed method and extensive experiments demonstrate the effectiveness of the proposed method
- the proposed method achieves reasonable performance improvements with negligible overhead
- the paper provided source codes in the supplementary material

**Weaknesses:**

- the paper adopts VBench for quantitative comparisons. however, the reliability of the VBench metrics is still not well justified. based on the reviewer's experience, some scores might favor specific aspects of videos while ignoring the actual visual quality. we can see some of the reported improvements are quite marginal
- the number of samples used in user study seems to be limited and how those samples are selected is not mentioned (randomly selected or not?)
- the experimental settings of inference efficiency seem to be unclear: how was the inference time measured? did the authors report a single run time or the averaged time of multiple inference runs
- in terms of long video generation, what if the proposed method is tested on HunyuanVideo+RIFLEx?

## other comments (not weaknesses)

- it might be slightly better to avoid using wrapfig

**Questions:**

please refer to the weaknesses section

---

### Note · Authors · 2025-11-14

I have read and agree with the venue's withdrawal policy on behalf of myself and my co-authors.